# Change Detection of Soil Formation Rate in Space and Time Based on Multi Source Data and Geospatial Analysis Techniques

**Qin Li** [1,2,3], **Shijie Wang** [1,3], **Xiaoyong Bai** [1,4,5,*] 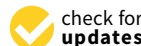, **Guangjie Luo** [5], **Xiaoqing Song** [6], **Yichao Tian** [7], **Zeyin Hu** [1,2,3], **Yujie Yang** [1,3,8] and **Shiqi Tian** [1,3,8]

1  Sate Key Laboratory of Environmental Geochemistry, Institute of Geochemistry, Chinese Academy of Sciences, Guiyang 550081, China; 17010090247@gznu.edu.cn (Q.L.); wangshijie@vip.skleg.cn (S.W.); huzeyin@mail.gyig.ac.cn (Z.H.); 17010090255@gznu.edu.cn (Y.Y.); 16010090310@gznu.edu.cn (S.T.)
2  University of Chinese Academy of Sciences, Beijing 100049, China
3  Puding Karst Ecosystem Observation and Research Station, Chinese Academy of Sciences, Puding 562100, China
4  CAS Center for Excellence in Quaternary Science and Global Change, Xi'an 710061, China
5  Guizhou Provincial Key Laboratory of Geographic State Monitoring of Watershed, Guizhou Education University, Guiyang 550018, China; luoguangjie@gznc.edu.cn
6  Geo-Engineering Investigation Institute of Guizhou Province, Guiyang 550008, China; 18010090240@gznu.edu.cn
7  School of Resources and Environment, Beibu Gulf University, Guangxi 530000, China; 18010090248@gznu.edu.cn
8  School of Geography and Environmental Sciences, Guizhou Normal University, Guiyang 550001, China
*  Correspondence: baixiaoyong@vip.skleg.cn; Tel.: +86-180-8509-9080

**Abstract:** Spatialization of soil formation rate (*SFR*) is always a difficult problem in soil genesis. In this study, the dissolution rate in karst areas of China during the period 1983–2015 was estimated on the basis of geospatial analysis techniques and detection of variation via the law of chemical thermodynamics in conjunction with long-term serial ecohydrology data. *SFR* at different lithological backgrounds was calculated on the basis of the content of acid-insoluble substances. Results showed that the spatial dissolution rate of carbonate rock ranges between 0 and 106 mm/ka, averaged at 22.51 mm/ka, and the *SFR* ranges between 10 and 134.93 t km$^{-2}$ yr$^{-1}$, averaged at 18.59 t km$^{-2}$ yr$^{-1}$. The dissolution rate and *SFR* exhibit a slight increasing trend with 0.04 mm/ka and 0.003 t km$^{-2}$ yr$^{-1}$, respectively. The risk for soil erosion was reevaluated on the basis of the *SFR* results, and the area with erosion risk and the ecologically safe area were corrected. Results indicated that the area with erosion risk is four times higher than the ecologically safe area. This study will hopefully instigate and facilitate the application and popularization of geospatial analysis technology to the research field of rock weathering and soil formation.

**Keywords:** karst; soil formation rate; carbonate; soil erosion; dissolution rate

## 1. Introduction

Approximately 25% of the world's population currently depends on the karst ecosystem for survival [1]. The environment of a karst landscape is fairly vulnerable [2–6], and in many karst areas, soil erosion and degradation have been regarded as severe geological disasters [7–10]. However, soil formation rate (SFR) is the key factor and important link restricting the formation and evolution of the karst ecosystem. Therefore, an in-depth study of the weathering of karst rocks and the formation and evolution of soil will promote the study and understanding of regional ecological risk assessment.

In the past several decades, a large scientific literature has reported on this subject of SFR. Morgan [11] believed that the mean *SFR* of source rock is ca. 100 t km$^{-2}$ yr$^{-1}$, varying from 10 t km$^{-2}$ yr$^{-1}$ to 300 t km$^{-2}$ yr$^{-1}$ depending on climatic and other factors. Norman Hudson [12], a soil conservation scientist, indicated that the mean *SFR* is approximately 1 cm/12a (equal to 11. 2 t km$^{-2}$yr$^{-1}$) under an ideal soil management condition. Wei QF [13] calculated the *SFR* in karst areas of south China and found that developing 1 cm of lime soil takes 13–32 thousand years, with a similar *SFR* to that in Yugoslavia wherein developing the same thickness of lime soil takes 20 thousand years. Yuan DX [14] determined that the time for carbonate rock to develop 1 cm of soil is approximately 2.5–8.5 thousand years. Cai ZX [15] reported that the *SFR* in karst areas of Guangxi is approximately 68–143 t km$^{-2}$ yr$^{-1}$ (approximately 11.6–24.4 thousand years) based on the dissolution rate of carbonate rock, the properties during soil formation, and the proportion of rocks. YL [16] found that the *SFR* in Guizhou Province is 6.84–103.46 t km$^{-2}$ yr$^{-1}$.

At present, the *SFR* of carbonate rock in karst areas is calculated on the basis of the dissolution rate of carbonate rock at the experimental scale of positioned observation and the content of acid-insoluble substances [15–17]. Long-term positioned observation only solves problems at the point or section scale. This approach requires considerable manpower, material resources, and financial resources; it is not applicable to large-scale continuous monitoring nor feasible for solving problems at the national or regional scale. Conventional approaches have constrained studies concerning the *SFR* of carbonate rocks and related scientific problems. Therefore, *SFR* at the spatial scale remains to be addressed. Remote sensing (RS) technology has played a huge role in the study of geoscientist scale extension [18–23]. Long-term, dynamic, and continuous large-scale data are provided by RS technology [24], e.g., precipitation (P), evapotranspiration (ET), and temperature (T) [25–27]. ZL [28] summarized the basic climatic data of relevant areas, such as T, P, and ET, to calculate the intensity of karstification.

In the present study, the main purposes are as follows: (1) Hydrochemical monitoring data of 43 basins and long time series of ecological and hydrological grid data were used to define the spatiotemporal difference of the dissolution rate of carbonate rocks via maximum potential dissolution (MPD) method and Multiple Linear Regression (MLR). (2) The dissolution rate, content of acid-insoluble substances, bulk weight, and content of carbonate rocks, as well as the *SFR* of noncarbonate rocks, were utilized to determine the *SFR* in rock weathering and calculate the temporal variation and spatial pattern of the *SFR* of carbonate rocks. (3) The effects of climate change on *SFR* under different lithologies were examined.

## 2. Materials and Methods

### 2.1. Study Area

China is one of the largest countries covered by karst landscape, which explains $3.44 \times 10^6$ km$^2$ and more than 1/3 of the total land area of China. The exposed area of carbonate rock is about $9.07 \times 10^5$ km$^2$, accounting for 1/7 of the total land area of China. The karst landscape is mainly distributed in eight provinces and regions in China, including Yunnan, Guizhou, Guangxi, Guangdong, Hunan, Hubei, Chongqing, and Sichuan Province. [29]. In karst areas, the longitudinal and latitudinal continuous distribution of heat and water is broken by the difference in geological structure and lithology. The weathered layer and degree of soil erosion are both influenced strongly by chemical reaction and hydropower, leading to a mosaic distribution of regional and non-regional soils. The vertical mosaic distribution of different types and different thickness of soils lead to a three-dimension highly spatiotemporal heterogeneity of karst landscape and weathering crust. Meanwhile, rocks are exposed and soil distributes discretely at the horizontal direction. At the vertical direction, the proportion of soil, rock and vegetation are different [30]. Classified by regions, karst areas can be divided into south karst area, north karst area and plateau karst area. Classified by lithology, it can be divided into homogenous carbonate rock (HC), carbonate rock intercalated with clastic rock (CI) and

carbonate/clastic rock alternations (CA). The HC is further divided into homogenous limestone (HL), homogenous dolomite (HD) and mixed dolomite/limestone (HDL) (Figure 1) [31].

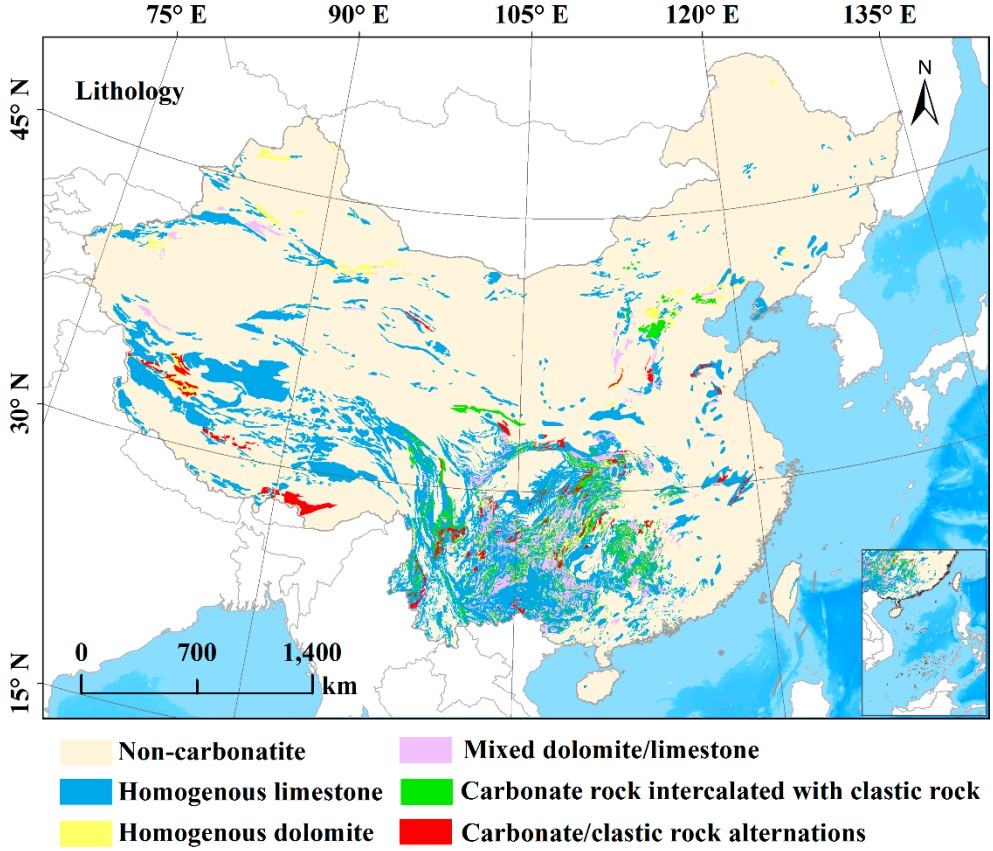

**Figure 1.** Lithologic assemblage of carbonate rocks in China.

## 2.2. Data Source and Pretreatment

This research involved a large amount of not only measured data but also ecological hydrological grid data, such as P, T, ET, soil moisture (SM), and NDVI. Table 1 described the data source, time span, and spatial resolution. For data of P, T, ET and SM, we accumulated monthly data of P, T, ET and SM into annual data of P, T, ET and SM, respectively. The *NDVI* dataset was a 15 days' composite value, First, we applied the maximum value composite (MVC) technique to synthesize monthly data [32], Second, we calculated the mean value of *NDVI* at 12 months per year to get the average *NDVI* per year. After that, all raster data were resampled to preserve the 0.083° spatial resolution of the *NDVI* data using a nearest neighbor algorithm replicating the pixels without changing the original cell values [33].

### 2.2.1. P, T, and ET

P, T, and ET data (1983–2015) were obtained from the Global Land Data Assimilation System (GLDAS) model at a resolution of 0.25° × 0.25° (https://ldas.gsfc.nasa.gov/gldas/). The goal of the NASA GLDAS was to ingest satellite- and ground-based observational data products by using advanced land surface modeling and data assimilation techniques for generating the optimal fields of land surface states and fluxes [34,35].

**Table 1.** Materials sources information.

| Parameters | Time Span | Temporal Resolution | Spatial Resolution | Sources |
|---|---|---|---|---|
| Precipitation (P) | 1983.1–2015.12 | Monthly | $0.25° \times 0.25°$ | Global Land Data Assimilation System |
| Temperature (T) | 1983.1–2015.12 | Monthly | $0.25° \times 0.25°$ | Global Land Data Assimilation System |
| Evaporation (ET) | 1983.1–2015.12 | Monthly | $0.25° \times 0.25°$ | Global Land Data Assimilation System |
| soil moisture (SM) | 1983.1–2015.12 | Monthly | $0.125° \times 0.125°$ | European Centre for Medium-Range Weather Forecasts |
| NDVI | 1983.1–2015.12 | 15 days | $0.083° \times 0.083°$ | Global Inventory Modeling and Mapping Studies |
| $Ca^{2+}$, $Mg^{2+}$, $Na^+$, $K^+$, $CO_3^{2-}$, $HCO_3^-$, $SO_4^{2-}$ and $Cl^-$ concentration | | multi-year average | / | GEMS-GLORI world river discharge database |
| Region boundaries | Present | / | / | State Bureau of China's Survey and Measurement. |
| Carbonate rock outcrops | Present | / | / | the Ministry of China Geological Survey |
| Soil erosion modulus | 1980s, 1990s, 2000, 2005, 2010, 2015 | Year | 1 km $\times$ 1 km | Institute of Mountain Hazards and Environment, Chinese Academy of Science. |

### 2.2.2. SM

The monthly SM dataset (1983–2015) [36] (https://www.ecmwf.int/) was acquired from the ERA-Interim reanalysis data of the European Centre for Medium-Range Weather Forecasts with a spatial resolution of 0.125°. The land Surface process used by the ERA-Interim is still TESSEL (Tiled ECMWF Scheme for Surface Exchanges over land). The dataset provided four layers of SM with soil depths of 7, 28, 100, and 289 cm, respectively. In this study, soil water data with a soil depth of 7 cm were selected.

### 2.2.3. NDVI

The *NDVI* data of the Global Inventory Modeling and Mapping Studies represent a vegetation product obtained by the AVHRR sensor mounted on a NOAA satellite [37,38]. The dataset had a spatial resolution of 8 km $\times$ 8 km (0.083°) and composite values of 15 days from 1982 to present [39]. The data were accessible online at https://ecocast.arc.nasa.gov/data/pub/gimms/. In this study, the time span of 1983–2015 was adopted.

### 2.2.4. Hydrochemical Data

Concentration data of $Ca^{2+}$, $Mg^{2+}$, $Na^+$, $K^+$, $CO_3^{2-}$, $HCO_3^-$, $SO_4^{2-}$, and $Cl^-$ were obtained from the GEMS-GLORI world river discharge database [40], which recorded information of more than

500 rivers. We selected hydrochemical data from 43 river basins in the database to calculate the ionic activity coefficients of calcium and bicarbonate ions (Figure 2). The multiyear average data were from 1996 to 2012.

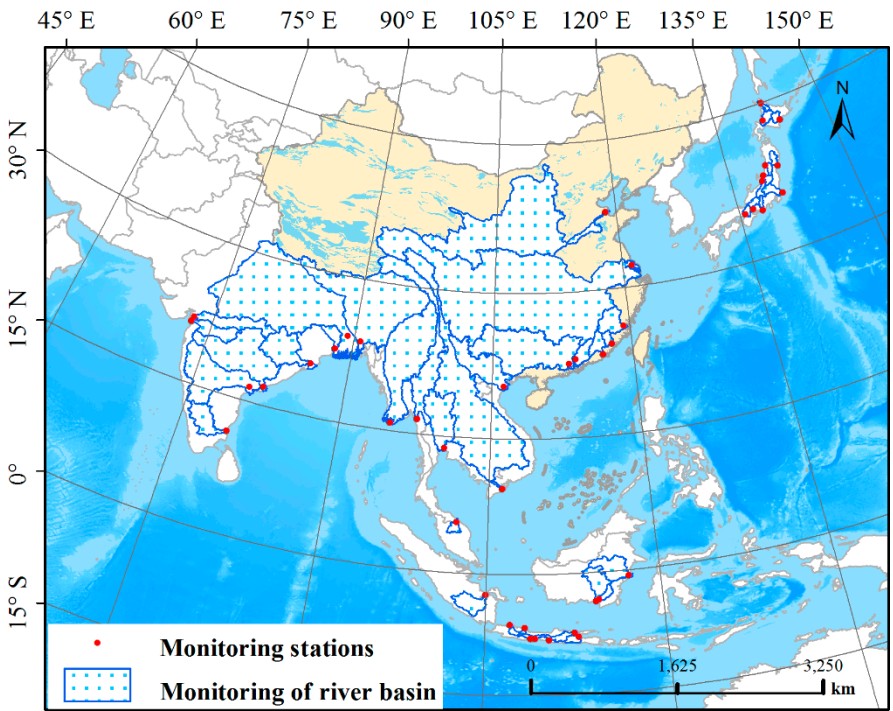

**Figure 2.** Distribution of watershed monitoring stations.

### 2.2.5. Carbonate Rock Outcrops

The distribution and zoning of carbonate rocks were from the geological map (1:500,000) and the map of soluble rock types in China (1:4,000,000) supplied by the Ministry of China Geological Survey. After digitalization, carbonate rocks in the karst areas of China were divided into HC, CI, and CA. Homogenous carbonate rocks were further divided into HL, HD, and HDL.

### 2.2.6. Soil Erosion Modulus

The soil erosion data (http://ir.imde.ac.cn/) were provided by the Institute of Mountain Hazards and Environment, Chinese Academy of Science. We utilized soil erosion modulus data for the time series of the 1980s and 1990s and 2000, 2005, 2010, and 2015 with a spatial resolution of 1 km [41].These data were calculated using the improved USLE model based on geographical big data. Equation (1) presented the basic form.

$$A = R \times K \times L \times S \times C \times P \times M \tag{1}$$

where $A$ (t·km$^{-2}$ yr$^{-1}$) refers to the amount of soil loss per unit area. $R$ (MJ·mm·hm$^{-2}$ yr$^{-1}$) refers to the rainfall erosivity factor. $K$ [t·hm$^2$·h/(hm$^2$·MJ·mm)] refers to the soil erodibility factor. $L$ refers to the slope length factor (dimensionless, 0–1), and $S$ refers to the slope factor (dimensionless, 0–1). $C$ refers to the coverage factor for vegetation. $P$ refers to the conservation measure factor, which included engineering and tillage measure factors. $M$ was a correction factor (dimensionless).

### 2.3. Methods

The objective is to solve the spatialization problem of SFR. On the basis of the law of chemical thermodynamics and through various platforms, such as ArcGIS, ENVI, $R$ studio, and Python, we must calculate the various parameters required for the dissolution rate. These parameters included

$Ca^{2+}$ activity coefficient ($\gamma_{Ca^{2+}}$) and $HCO^{3-}$ activity coefficient ($\gamma_{HCO_3^-}$), limestone solubility product constant ($K_s$), equilibrium constant of $CO_2$ dissolved in water ($K_0$), first-order ionization constant of $H_2CO_3$ ($K_1$), second-order ionization constant of $H_2CO_3$ ($K_2$), and carbon dioxide partial pressure ($pCO_2$). (1) In order to simulate the spatial distribution of activity coefficients of $Ca^{2+}$ and $HCO_3^-$ ions, firstly, the $Ca^{2+}$ and $HCO_3^-$ activity coefficients of 43 stations were calculated by using the hydrochemical data of 43 river basins. Then the corresponding values of P, ET, T, *NDVI* and SM were extracted from the geographical coordinates of 43 stations. The activity coefficients of $Ca^{2+}$ and $HCO_3^-$ ions were used as dependent variables, and the extracted P, ET, T, *NDVI* and SM values were used as independent variables. Multiple linear regression (MLR) was carried out in SPSS, and the relationship between the dependent variables and independent variables was quantified. Finally, the spatial distribution of $Ca^{2+}$, $HCO_3^-$ ion activity coefficients is retrieved by the regression relationship and P, ET, T, *NDVI* and SM spatial data. (2) $K_s$, $K_0$, $K_1$, and $K_2$ were calculated by T. (3) $pCO_2$ was measured by ET. We calculated the dissolution rate of limestone via the MPD method. Then, we obtained the spatial distribution of five lithological dissolution rates in the karst areas of China on the basis of the dissolution ratio of other lithologies and limestones. Finally, we measured the *SFR* by combining the density of rocks and the content of acid-insoluble substances (Figure 3).

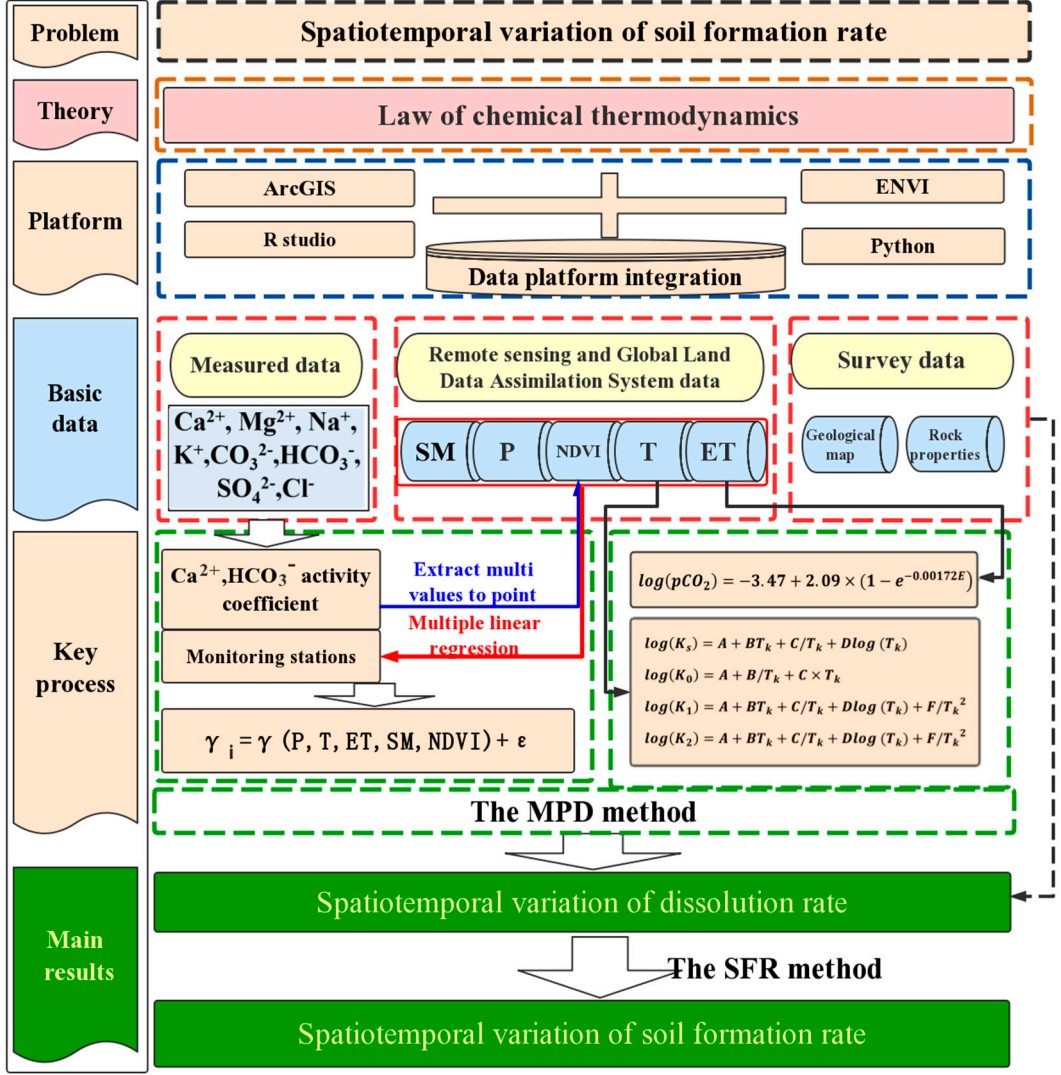

**Figure 3.** Technical flowchart.

### 2.3.1. Maximal Potential Dissolution (MPD) Method

We used the MPD method to calculate the dissolution rate of limestone in China, which had been applied in quantifying karstification at global and regional scales [42,43]. The intensity of karstification can be obtained by the MPD method if the temperature, precipitation and evapotranspiration of a given area are known. The formula is expressed as follows:

Based on the usual equations for the dissolution of calcite at equilibrium given in Equation (2), Gombert created the MPD method of Equation (3) for carbonate zones with the assumption that the chemical reactions were at equilibrium in local hydrological, meteorological and geochemical conditions [42–45].

$$CaCO_3 + CO_2 + H_2O \rightleftharpoons Ca^{2+} + 2HCO_3^- \tag{2}$$

$$D_{max} = 10^6 (P - E)\left[Ca^{2+}\right]_{eq} = 10^6 (P - E)\left(K_s K_1 K_0 / 4K_2 \gamma_{Ca^{2+}} \gamma_{(HCO_3^-)}^2\right)^{1/3} (pCO_2)^{1/3} \tag{3}$$

where $D_{max}$ is the maximum potential dissolution rate (mol $km^{-2}yr^{-1}$), $P$ and $E$ are the total rainfall and evapotranspiration, respectively. $K_S$ is the calcite solubility constant, $K_1$ is the equilibrium constant of $CO_2$ hydration and dissociation as $HCO_3^-$, $K_0$ is the equilibrium constant of $CO_2$ in water, $K_2$ is the equilibrium constant for $CO_3^{2-}$. $\gamma_{Ca^{2+}}$ and $\gamma_{HCO_3^-}$ are the activity coefficients of $Ca^{2+}$ and $HCO_3^-$ ions in water. $pCO_2$ is the partial pressure of $CO_2$ in soil or aquifer.

Table 2 and Equations (4)–(7) showed that $K_s$, $K_1$, $K_0$ and $K_2$ were the functions of temperature $T_k$ (°K) [46].

$$log(K_s) = A + BT_k + C/T_k + Dlog(T_k) \tag{4}$$

$$log(K_1) = A + BT_k + C/T_k + Dlog(T_k) + F/T_k^2 \tag{5}$$

$$log(K_0) = A + B/T_k + C \times T_k \tag{6}$$

$$log(K_2) = A + BT_k + C/T_k + Dlog(T_k) + F/T_k^2 \tag{7}$$

**Table 2.** Factor coefficient of $K_i$.

|        | A         | B           | C         | D          | F         |
|--------|-----------|-------------|-----------|------------|-----------|
| $K_s$  | −171.9065 | −0.07793    | 2839.3191 | 71.595log  |           |
| $K_1$  | −356.3094 | −0.06091964 | 21834.37  | 126.8339   | −1684915  |
| $K_0$  | −14.0184  | 2385.73     | 0.015264  |            |           |
| $K_2$  | −107.8871 | −0.03252849 | 5151.79   | 38.92561   | −563713.9 |

The activities of $Ca^{2+}$ and $HCO_3^-$ were calculated according to the formula below.

$$log(\gamma_i) = -AZ_i^2 \frac{\sqrt{I}}{1 + Ba_i \sqrt{I}} \tag{8}$$

$A$ and $B$ are the coefficients related to temperature. The calculation formula was as follows:

$$A = 0.4883 + 8.074 \times 10^{-4} T \tag{9}$$

$$B = 0.3241 + 1.6 \times 10^{-4} T \tag{10}$$

$I$ is the strength of solution ions. The formula was as follows:

$$I = \frac{1}{2} \sum_i Z_i^2 C_i \tag{11}$$

where $C_i$ is the ionic concentration (mol·$L^{-1}$), $Z_i$ is the charge of the ionic.

The partial pressure of $CO_2$ in soil or water-bearing stratum was calculated in accordance with Brook's formula, as follows [47].

$$log(pCO_2) = -3.47 + 2.09 \times \left(1 - e^{-0.00172E}\right) \tag{12}$$

The above calculation is the maximum dissolution rate of limestone, and the difference of dissolution rate of different lithology can be characterized by dissolution coefficient (DC). The calculation formula of dissolution rate of carbonate rocks of various lithologies was as follows [48]:

$$v = DC \times Dmax \tag{13}$$

When estimating the dissolution rate of carbonate rock, the specific degree of different lithology was considered as the coefficient of lithology (Table 3).

**Table 3.** Dissolution coefficient of various lithologic assemblages.

| Lithology | HL | HD | HDL | CI | CA |
|:---:|:---:|:---:|:---:|:---:|:---:|
| DC | 0.965 | 0.505 | 0.8015 | 0.767 | 0.767 |

### 2.3.2. Method of Computing SFR

In previous studies, the average dissolution rate was often used to replace the regional dissolution rate. However, the dissolution rate was closely related to temperature, precipitation, hydrological and other environmental conditions, different regions had different dissolution rate. therefore, in this paper, the dissolution rate of pixel scale had calculated by MPD method was substituted into the Equation (14) to reflect the spatiotemporal heterogeneity of soil formation rate [16,31]. As per the results of an in-house laboratory investigation, The density of limestone and dolomite is 2.75 and 2.86 t m$^{-3}$. The soil formation rate of other rock types is 200 t km$^{-2}$ yr$^{-1}$. The content of acid-insoluble components is shown in Table 4.

$$SFR = vQ\rho C + R(1 - C) \tag{14}$$

where *SFR* is the soil formation rate (t km$^{-2}$ yr$^{-1}$), *v* is the dissolution velocity of carbonate rocks (m$^3$km$^{-2}$ yr$^{-1}$), *Q* is the content of acid-insoluble components (%), $\rho$ is the carbonate density (t m$^{-3}$), *C* is the proportion of carbonate and *R* was the soil formation rate of other rock types. The acid-insoluble components can be computed based on 5, 20 and 50% for HC, CI and CA. In addition, carbonate rock can be computed based on 95, 80 and 50% for HC, CI and CA (Table 4).

**Table 4.** Characteristics of various lithologic assemblages.

| Lithology | | C | Q |
|:---:|:---:|:---:|:---:|
| HC | HL<br>HD<br>HDL | >90% | <10% |
| | CI | 70~90% | 10~30% |
| | CA | 30~70% | 30~70% |

The acid-insoluble components can be computed based on 5, 20 and 50% for HC, CI and CA. In addition, carbonate rock can be computed based on 95, 80 and 50% for HC, CI and CA.

### 2.3.3. Multiple Linear Regression (MLR)

MLR was mainly used to study the correlation between a dependent variable and multiple independent variables [49–51], and had a wide range of applications. The basic structure of multiple linear regression model was as follows:

$$y_a = b_0 + b_1 x_{1a} + b_2 x_{2a} + \ldots + b_k x_{ka} + \varepsilon_a \tag{15}$$

where $y_a$ is dependent variable, $x_{1a}, x_{2a}, \ldots, x_{ka}$ are independent variables, $b_0, b_1, \ldots, b_k$ are undetermined parameter. $\varepsilon_a$ is random variable. In this study, the dependent variable is the ion activity coefficient, and the independent variables are *P*, *T*, *ET*, *SM* and *NDVI*.

### 2.3.4. Least Squares Trend Analysis

We perform the least squares linear regression analysis to quantify the trend of SFR. This method has been extensively performed at regional to global scales as a means of understanding the evolution trend of time-series data [52,53]. The formula is expressed as follows:

$$K = \frac{n \sum_{i=1}^{n} (i \times M_{SFR,i}) - \sum_{i=1}^{n} i \times \sum_{i=1}^{n} M_{SFR,i}}{n \times \sum_{i=1}^{n} i^2 - \left(\sum_{i=1}^{n} i\right)^2} \tag{16}$$

where the value of n is 33, $i$ is the year number, $M_{SFR}$ is the average value of the *SFR*, and $i$ is the *SFR* of the *i* th year. *K* is the regression slope. When $K > 0$, the evolution trend of *SFR* increases in the 33-year period; otherwise, this trend decreases. No uniform standard exists for the division of *K*-value trends [54,55]. According to the overall distribution of the *SFR* in the study area and the calculation of the changes in the *K*-value, the *K*-value conforms to a normal distribution [56]. Thus, *SFR* is used in the equal-pitch division method. We divide the *K*-value into seven levels, namely, significant decrease ($K \leq -0.05$), slight decrease ($-0.05 < K \leq -0.01$), constant ($-0.001 < K \leq 0.01$), slight increase ($0.01 < K \leq 0.05$) and significant increase ($K \geq -0.05$).

### 2.3.5. Pearson Correlation Analysis

For more than a decade, scholars had used *SFR* data from different regions, time series, and resolutions to study the relationship among variables [57,58]. The determination of the relationship between *SFR* and climatic factors is mainly accomplished by calculating and verifying the correlation coefficients [59]. The formula is presented as follows:

$$R_{xy} = \frac{\sum_{i=1}^{n} \left[\left(x_i - \overline{X}\right)\left(y_i - \overline{Y}\right)\right]}{\sqrt{\sum_{i=1}^{n} \left(x_i - \overline{X}\right)^2 \sum_{i=1}^{n} \left(y_i - \overline{Y}\right)^2}} \tag{17}$$

where $n$ is the number of samples; $X^-$ and $Y^-$ are the means of variables $x$ and $y$, respectively; and $R_{xy}$ is the correlation coefficient between variables x and y. If $|R| \leq 0.5$, then the correlation between *SFR* and climatic factors is insignificant; if $|R| \geq 0.5$, then the correlation coefficients are taken as statistically significant at $p = 0.05$.

## 3. Results

### 3.1. Spatialization of Activity Coefficients of Calcium Ions and Bicarbonate

We selected the hydrochemical data of 43 research areas and nearby monitoring stations from the GEMS-GLORI world river discharge database [40] and calculated the $\gamma_{Ca^{2+}}$ and $\gamma_{HCO_3^-}$ of 43 monitoring stations. Then, the corresponding values of *P*, *ET*, *T*, *NDVI*, and *SM* grids were obtained through the geographical coordinates of the 43 monitoring stations. Subsequently, the two

ion activity coefficients were regressed by MLR with *P*, *ET*, *T*, *NDVI*, and *SM*. The results showed that the MLR equations were effective, with $R^2$ reaching 0.79 (*p* <0.01) (Equation (18)). Table 5 lists the regression coefficients.

$$\gamma_i = \gamma(P, T, ET, SM, NDVI) + \varepsilon$$
$$= (b_0 + b_1 P + b_2 T + b_3 ET + b_4 SM + b_5 NDVI) \tag{18}$$
$$+ \varepsilon(R2 = 0.7909, \; R2 = 0.7914)$$

**Table 5.** The coefficient of $\gamma_{Ca^{2+}}$ and $\gamma_{HCO_3^-}$ fitted with MLR.

|  | $b_0$ | $b_1$ | $b_2$ | $b_3$ | $b_4$ | $b_5$ | $\varepsilon$ |
|---|---|---|---|---|---|---|---|
| $\gamma_{Ca^{2+}}$ | 0.8296 | $5.8 \times 10^{-7}$ | 0.0003 | $1.2713 \times 10^{-6}$ | $-3.7831 \times 10^{-5}$ | 0.0068 | 0.0002 |
| $\gamma_{HCO_3^-}$ | 0.9413 | $2.15 \times 10^{-7}$ | $-1.20 \times 10^{-4}$ | $4.72 \times 10^{-7}$ | $-1.40 \times 10^{-5}$ | $-0.0025$ | $-0.0001$ |

In Figure 4, the spatial distribution of $\gamma_{Ca^{2+}}$ and $\gamma_{HCO_3^-}$ was high in northwest and low in southeast; the value ranges were 0.818–0.833 and 0.936–0.942, with averages of 0.823 and 0.939, respectively. The spatial distribution pattern of the two ion activity coefficients is similar because $Ca^{2+}$ and $HCO_3^-$ have high correlation.

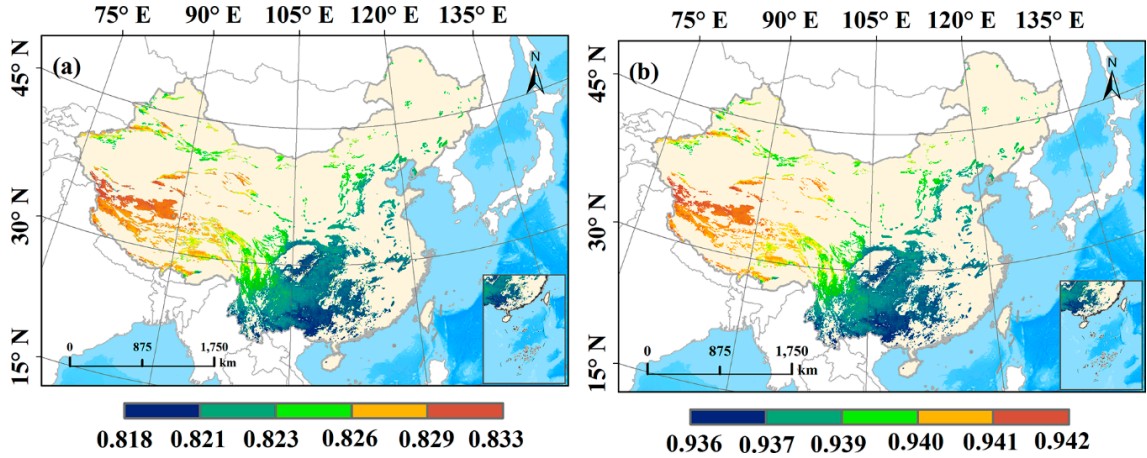

**Figure 4.** The multi-year mean spatial distribution of the activity coefficients of $Ca^{2+}$ (**a**) and $HCO_3^-$ (**b**).

*3.2. Diversity of Dissolution Rate and Its Evolutionary Rule*

3.2.1. Spatial Pattern of Dissolution Rate

Influenced by landform, topography, and meteorological factors, the dissolution rate is high in the southeast and low in the northwest. It also showed a significant regional difference. On the basis of the spatial distribution during the period 1983–2015 (Figure 5a), the dissolution rate of the study area ranged between 0 mm/ka and 106 mm/ka, averaged at 22.51 mm. Spatially, the dissolution rate is high in southeast and low in northwest. The mean values of different grades of dissolution rate were analyzed statistically. The results showed that 19.40% of the study area had a dissolution rate of <10 mm/ka, distributed mainly in the northwest part. Moreover, 29.50% of the study area had a dissolution rate of 10–30 mm/ka, distributed mainly in the west and east parts, and 37.71% had a dissolution rate of 30–50 mm/ka, distributed mainly in the central south part. Furthermore, 5.72% of the study area had a dissolution rate of 50–60 mm/ka, distributed mainly in the southeast part, and 7.67% had a dissolution rate of >60mm/ka, distributed mainly in the southeast coastal area.

As displayed in Figure 5b, the dissolution rate decreases with the increase of latitude. The dissolution rate at different latitudes fluctuated considerably during 1983–2015, with four high-value areas. In particular, the area with the highest fluctuation was located at 13.66° N with a maximal

dissolution rate of 94 mm/ka. The area with the second highest value was located at 21.50° N–22.50° N, which is the most active area of karstification in China, with a maximal dissolution rate of 64 mm/ka. The area with the third highest value was located in central south China (27.44° N–28.50° N), with a maximal dissolution rate of 33.70 mm/ka. The fourth area with high dissolution rate was located at 42.04° N–43.79° N, with a maximal dissolution rate of 19.34 mm/ka. As the south area was more easily affected by monsoon than the north area, considerable rainfall and numerous thermal resources were observed. Therefore, the dissolution rate in the south area fluctuated greater than that in the north area.

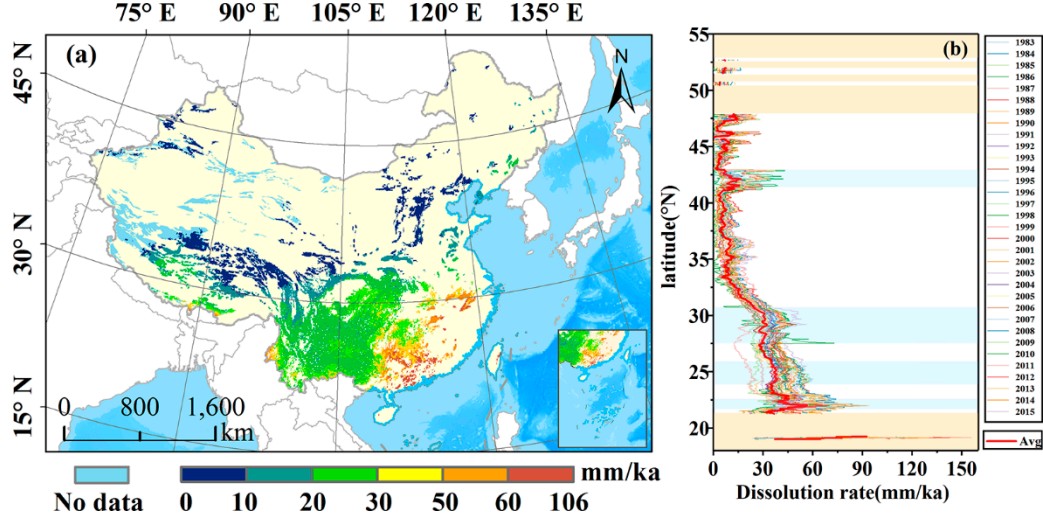

**Figure 5.** Spatial distribution (**a**) and latitudinal variation (**b**) of average annual dissolution rate of karst in China during 1983~2015.

### 3.2.2. Evolutionary Process of Dissolution Rate

As shown in Figure 6b, the change of the dissolution rate from 1983 to 2007 was relatively stable, while the change of the dissolution rate from 2008 to 2015 was relatively drastic. In general, the dissolution rate showed a slight upward trend, with an acceleration rate of 0.04 mm/ka during 1983–2015. In particular, the lowest dissolution rate (21.96 mm/ka) in the study area occurred in 2009, and the highest (33.92 mm/ka) occurred in 2015.

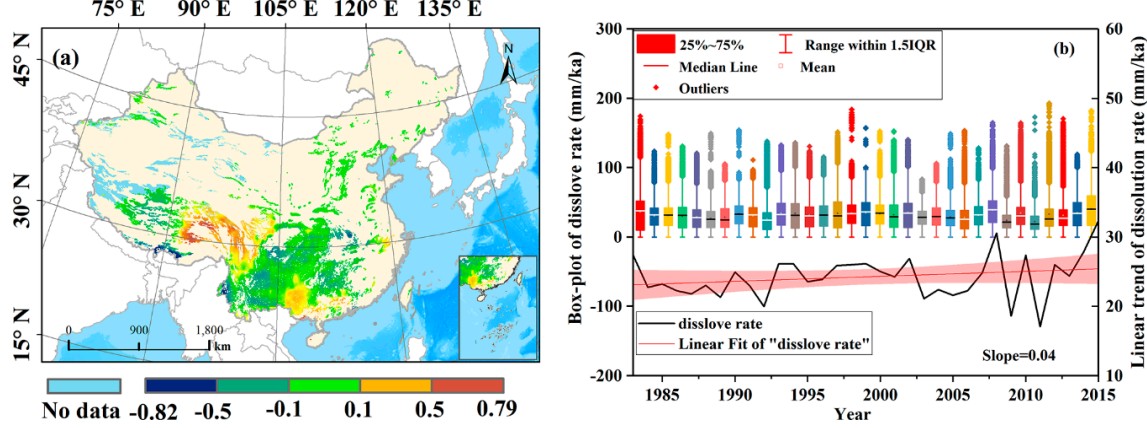

**Figure 6.** The changing trend of dissolution rate (**a**) and annual variation of dissolution rate (**b**).

To understand the spatiotemporal evolutionary pattern of the dissolution rate fully, the evolutionary trend during 1983–2015 was analyzed at the pixel scale (Figure 6a). The results showed that the dissolution rate in the southwest area tended to decrease, whereas that in the south area tended

to increase. The proportions of significant reduction, slight reduction, no change, slight increase, and significant increase were 6.96%, 32.46%, 35.73%, 16.02%, and 8.83%, respectively. As illustrated in Figure 6a, significant reduction was sporadically distributed in the south area, and slight reduction was mainly distributed in the southwest area. The area with slight increase was mainly situated at 27° N–30° N, and the significant increase was mainly distributed in the west and southeast areas. The area with no change was only sporadically distributed.

### 3.3. Diversity of SFR and Its Evolutionary Process

#### 3.3.1. Spatial Pattern of SFR

Influenced by lithology, landform, topography, and meteorological factors, *SFR* in the study area showed a significant regional difference. On the basis of the multiyear average *SFR* during 1983–2015, the *SFR* in the study area ranged between 10 t km$^{-2}$ yr$^{-1}$ and 134.93 t km$^{-2}$ yr$^{-1}$, averaged at 18.59 t km$^{-2}$ yr$^{-1}$ (Figure 7a). Spatially, the *SFR* was high in southeast and low in northwest. The different grades of *SFR* were analyzed statistically. The results showed that 86.44% of the study area had an *SFR* of <15 t km$^{-2}$ yr$^{-1}$, distributed anywhere in the study area. Furthermore, 9.06% of the study area had an *SFR* ranging between 15 t km$^{-2}$ yr$^{-1}$ and 55 t km$^{-2}$ yr$^{-1}$, distributed sporadically, and 4.50% had an *SFR* ranging between 55 t km$^{-2}$ yr$^{-1}$ and 134.93 t km$^{-2}$ yr$^{-1}$, distributed dispersively.

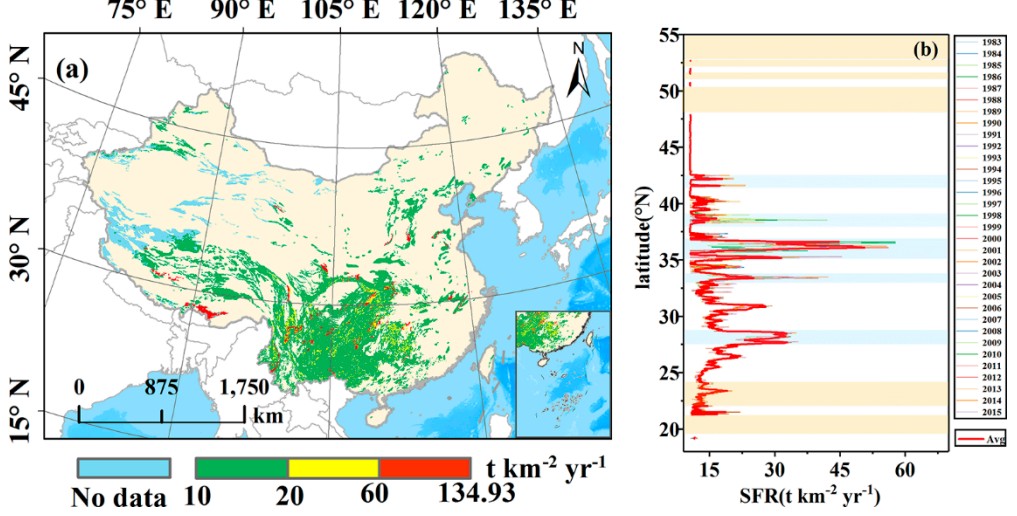

**Figure 7.** Spatial distribution of average annual *SFR* (**a**) and latitudinal distribution of mean value (**b**) in the karst area of China during 1983~2015.

As presented in Figure 7b, the *SFR* at different latitudes fluctuated considerably during 1983–2015, with three high-value areas. In particular, the highest *SFR* occurred at 36.54° N, with a maximal dissolution rate of 56.92 t km$^{-2}$ yr$^{-1}$. The second highest *SFR* occurred at 27° N–28° N, which is the most active area of karstification in China, i.e., south karst area, with a maximal dissolution rate of 36 t km$^{-2}$ yr$^{-1}$. The one-to-one correspondence between the *SFR* and dissolution rate was not observed, and this phenomenon was closely related to that of the lithology.

#### 3.3.2. Dynamic Variation of SFR

As shown in Figure 8b, the *SFR* in the study area fluctuated upward at an accelerating rate of 0.003 mm/ka during 1983–2015. In particular, three low-value and three high-value periods were observed. The periods with low values were years 1987, 2003, and 2009, with *SFRs* of 18.35, 18.39, and 18.40 t km$^{-2}$ yr$^{-1}$, respectively. The periods with high values were years 1984, 1994, and 2015, with *SFRs* of 18.78, 18.80, and 18.77 t km$^{-2}$ yr$^{-1}$, respectively.

The evolutionary trend of *SFR* was analyzed at the pixel scale (Figure 8a). Generally, a slight increase in *SFR* occurred locally, whereas in most areas, it remained stable. On the basis of the area proportion of different *SFRs* (Figure 8a), the proportions of significant reduction, slight reduction, no change, slight increase, and significant increase were 4.54%, 0.06%, 75.03%, 19.14%, and 1.23%, respectively.

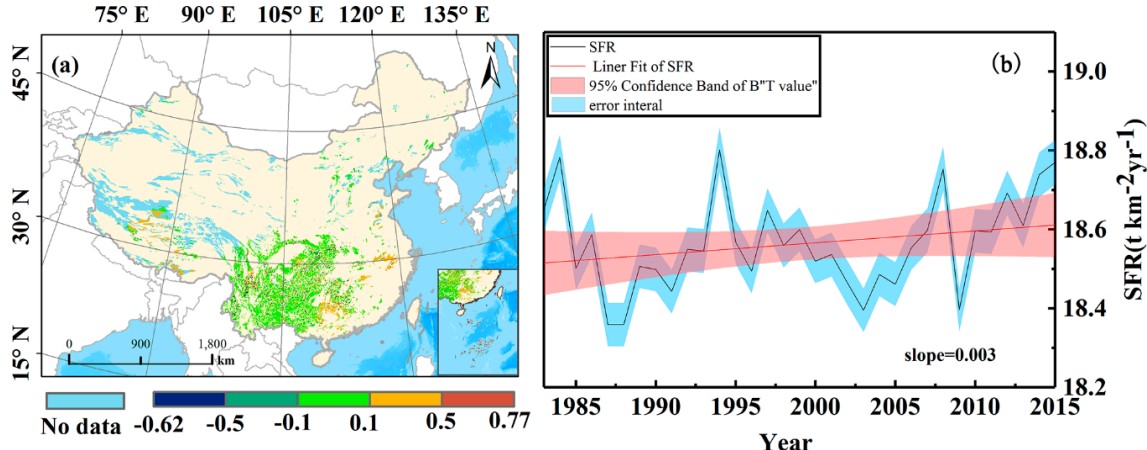

**Figure 8.** The evolutionary trend of *SFR* (**a**) and annual variation of *SFR* (**b**).

### 3.3.3. Statistical Characteristics of SFR under Different Lithologies

As shown in Table 6, the statistical results of multiyear *SFR* under different lithologies presented a trend of $SFR_{CA} > SFR_{CI} > SFR_{HC}$, and the *SFRs* of CA, CI, and HC were 111.13, 47.98, and 12.01 t·km$^{-2}$ yr$^{-1}$, respectively. In terms of HC, the trend was $SFR_{HL} > SFR_{HDL} > SFR_{HD}$, and the *SFRs* of HL, HDL, and HD were 12.25, 12.17, and 10.85 t·km$^{-2}$ yr$^{-1}$, respectively.

*SFR* has multiple values and diverse characteristics. The value of *SFR* is related to the content of acid-insoluble substances in rocks. When carbonate rock occurs in HC, CA, and CI, its *SFR* may differ considerably because the key factors affecting *SFR* are related to the contents of acid-insoluble substances in rocks, in addition to the contents of clastic rock in the stratum of a region.

**Table 6.** Statistic *SFR* of each lithology at pixel scale during 1983~2015.

| Lithology | | Mean | Min | Max | Std dev. |
|---|---|---|---|---|---|
| HC | | 12.01 | 10 | 20.71 | 1.45 |
| | HL | 12.25 | 10 | 20.71 | 1.49 |
| | HD | 10.85 | 10 | 12.92 | 0.78 |
| | HDL | 12.17 | 10.01 | 18.90 | 1.12 |
| CI | | 47.98 | 41.04 | 61.40 | 2.81 |
| CA | | 111.13 | 100 | 134.93 | 6.29 |

### 3.4. Correlation between Ecohydrological Factors and SFR under Different Lithological Backgrounds

Figure 9b–f display the correlations between $SFR_{HL}$, $SFR_{HD}$, $SFR_{HDL}$, $SFR_{CI}$, and $SFR_{CA}$ and the ecological hydrological factors, respectively. $SFR_{HL}$, $SFR_{HD}$, and $SFR_{HDL}$ had strong correlations with P, SM, and T, and all correlation coefficients exceeded 0.65. They had the strongest correlation with P, with correlation coefficients of 0.91, 0.94, and 0.92, respectively. The correlation coefficient of $SFR_{HD}$ and SM reached 0.92, whereas the correlation coefficients of $SFR_{HL}$ and $SFR_{HDL}$ and SM were only 0.71 and 0.65, respectively. The reason may be the better soil continuity of HD than that of HL and HDL, which increases the correlativity of $SFR_{HD}$ and SM.

Figure 9e,f clearly illustrate that the correlations between $SFR_{CI}$ and $SFR_{CA}$ and P were good, with correlation coefficients of 0.89 and 0.75, respectively, but the correlations between the two and SM,

T, and ET were poor. The main factor is that they contain a high proportion of acid-insoluble matter, which is the main source of weathered soil. Furthermore, the content of rock acid-insoluble matter is mainly affected by the diagenesis process and less effected by the outside world, leading $SFR_{CI}$ and $SFR_{CA}$ and all factors, except P, to respond poorly.

In conclusion, P is the most important climatic factor that affects the carbonate formation rate.

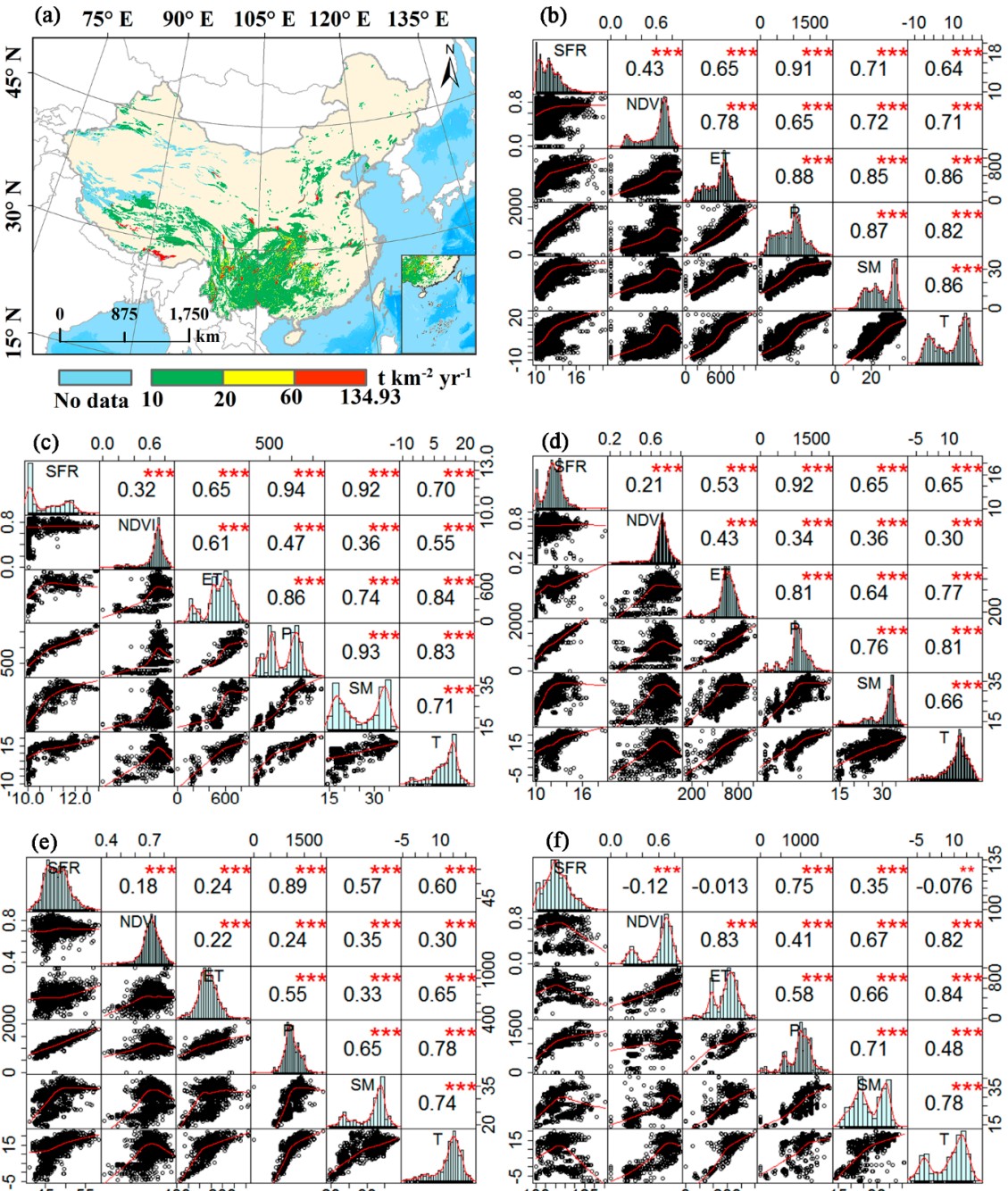

**Figure 9.** Correlation coefficient matrix between *SFR* of various lithology and ecohydrological factors: (**a**) Spatial pattern of average *SFR* from 1983 to 2015, (**b**) *SFR* of HL and ecohydrological factors. (**c**) *SFR* of HD and ecohydrological factors, (**d**) *SFR* of HDL and ecohydrological factors, (**e**) *SFR* of CI and ecohydrological factors, (**f**) *SFR* of CA and ecohydrological factors. ** indicates a significant correlation at the 0.01 level (both sides); *** indicates a significant correlation at the 0.001 level (both sides);

## 4. Discussion

*4.1. Comparison with Similar Studies*

4.1.1. Dissolution Rate

We compared our results with the relevant studies on the watershed scale [48,60–62], the spatial distribution map of the dissolution rate calculated in this study was clipped and compared with the vector boundary of relevant studies, and found that the results of this study were similar to those of many studies, although they are different from the existing research results, they could explicable. For example, Wang LC calculated the average dissolution rate of Muzhu cave watershed in Guizhou Province as 41.5 mm/ka by calcium ion concentration [60], which is 2.9 mm/ka higher than the result in this study (38.6 mm/ka). The reason is that the method of calculating dissolution rate by ion concentration does not exclude the influence of exogenous acid on dissolution. Han GL [61] and Zeng C [62] estimated the dissolution rate of Wujiang River and Banzhai watershed as 33 mm/ka and 24.91 mm/ka by hydro-chemical runoff method, which slightly deviated from the results of this study by 2.05 mm/ka and 2.31 mm/ka, respectively, this is because the method of solute load relies on manual monitoring and sampling test, and different sampling seasons and locations will affect the calculation results. Cao JH monitored the dissolution rate of limestone in the Pearl River watershed through the dissolution test piece method, and obtained that the dissolution rate of the Pearl River watershed is 21.4 mm–115.1 mm [63], the result of dissolution rate of the Pearl River watershed clipped in this paper is 16–101.32 mm, the test piece buried in the soil is susceptible to the influence of bio-organic acids or other exogenous acids in the soil, making the dissolution rate higher than the results in this study (Table 7).

**Table 7.** Compare with other results of dissolution rate.

| Source | Study Area | Lithology | Dissolution Rate (mm/ka) | This Study | Ratio, This to Other |
|---|---|---|---|---|---|
| Wang LC (2010) [60] | Muzhu cave watershed | HL | 41.5 | 38.6 | 2.9 |
| Cao JH (2011) [48] | Pearl River watershed | HL, HD, HDL, CI, CA | 21.4~115.1 | 16~101.32 | 5.4~13.78 |
| Zeng C (2017) [62] | Banzhai watershed in Province | HL | 24.91 | 27.22 | −2.31 |
| Han GL (2005) [61] | Wujiang | HL | 33 | 30.95 | 2.05 |
| This study (2019) | Carbonate area of China | HL, HD, HDL, CI, CA | 0~106 | | |

4.1.2. Soil Formation Rate

We compared the calculation results with related [16,31,63] studies and found that the results in this paper were similar to those of existing studies, but slightly higher than those of existing studies. For example, the soil formation rate calculated by Li YB 16] in Guizhou Province was between 6.75 to 103.46 t km$^{-2}$ yr$^{-1}$. The range of this article was from 10.78 to 117.68 t km$^{-2}$ yr$^{-1}$; the soil formation rate calculated by Cao JH [63] in southwest China was from 4 to 120 t km$^{-2}$ yr$^{-1}$, while the range calculated in this paper was from 10 to 122.77 t km$^{-2}$ yr$^{-1}$; the soil formation rate calculated by Li Y [31] in southern China was to serve the allowable soil loss, so only one estimate was proposed, and the estimation result was $20 < SFR \leq 100$. The reason for the higher results in this paper was that the maximum potential dissolution method was used in calculating the dissolution rate. In an open karst system, karstification may not reach the dissolution equilibrium, so the dissolution rate calculated in this study was the maximum. However, in previous studies, spatial heterogeneity was not considered in calculating soil formation rate, and the value of dissolution rate was one lithology and one value. Therefore, the soil formation rate calculated by previous studies will be lower than this study.

*4.2. Application of Soil Formation Rate in the Risk Reassessment of Soil Erosion*

In China, allowable amount of soil erosion is an indicator for the risk assessment of water and soil erosion. The Standards for Classification and Gradation of Soil Erosion indicated that the allowable amount of soil erosion in karst area is 500 t km$^{-2}$ yr$^{-1}$, and any areas with lower amount of soil erosion

are safe for soil erosion. The calculation of the present study shows that the average *SFR* in the karst areas in China is 18.59 t km$^{-2}$ yr$^{-1}$, which is equal to 3.7% of the allowable amount of soil erosion of 500 t. Therefore, the existing Standards for Classification and Gradation of Soil Erosion are far from stringent, thereby leading to a negligence of soil erosion risks in karst areas. This condition is probably also a reason for the occurrence and development of soil degradation and solidification in the karst areas. Therefore, considerable attention should be paid to soil conservation in the karst areas. In other words, low modulus of soil erosion does not necessarily mean low risks of soil erosion. Many field observations have shown that erosion in karst areas is low, but the risk of erosion is very high because of the slow soil formation rate, thin soil layers, and in some places even no soil can to lose [64,65]. Therefore, the risk of soil erosion should be determined by the relation of actual amount of soil erosion and *SFR* and not by the actual amount of erosion or the erosion modulus.

*SFR* is theoretically the upper limit of allowable amount of soil erosion in karst areas and can be the threshold minimum of soil erosion risk. If the theoretical amount of erosion is higher than the SFR, then the area will be in danger of soil erosion. If the theoretical amount of erosion is lower than the SFR, then the area will be safe for soil erosion. If the theoretical amount of erosion is equal to the SFR, then the area will be a critical area for soil erosion. Using the theoretical erosion rate supplied by the database of Mountain Hazards and Environment of China, we conclude that the safe area for soil erosion explains 20%, and the risky area explains 80% of the carbonate rock areas. The risky area is four times the safe areas (Figure 10). Therefore, considerable attention should be paid to soil conservation in this area.

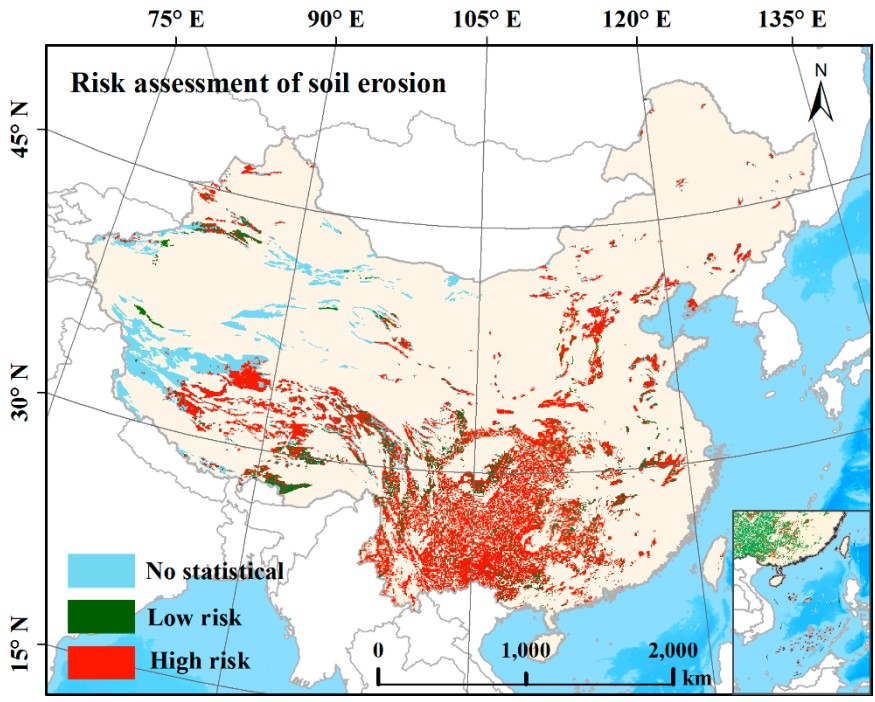

**Figure 10.** Risk assessment of soil erosion in the karst area of China.

### 4.3. Applicability of the Method

The method used in this study is divided into two parts. The first part is the MPD method to calculate the dissolution rate, and the second part is the method to calculate SFR.

The MPD method is applicable to all limestone areas [42,43,45]. In this study, we calculate the dissolution rate of carbonate rock on the basis of the specific corrodibility of other rocks in the carbonate rock and limestone. The parameters used in the method can be calculated from data such as T, P, and ET. These data comprise RS or reanalysis data products. The link in the model that requires the

measured data is a simulation of $Ca^{2+}$ and $HCO_3^-$ ion activity coefficients. When performing ion concentration simulation, $Ca^{2+}$ and $HCO_3^-$ ion activity coefficients are calculated, and an estimation model is established with local environment meteorological data. The meteorological and hydrological spatial data are inversely transformed into space to achieve the spatialization of the ion activity coefficient. The MPD method can also be extended to other soluble rock areas. This method can be generalized only through replacing the chemical equation, deriving a new formula on the basis of the principle of chemical thermodynamics, and changing the solubility product constant (Ks) of the rock in water.

In addition, the *SFR* model is mainly calculated on the basis of the carbonate rock dissolution velocity, rock density, acid insoluble matter, and the proportion of carbonate rock. Therefore, the model is only applicable to karst carbonate areas instead of all karst areas. In recent years, this model has been widely used in permissible amount of carbonate soil loss and its erosion characteristics in southwestern China [16,31].

In summary, although the method used in this study is only applicable to carbonate rock areas, the global carbonate rock area is as high as 22 million km$^2$ [66]. Approximately 25% of population depends on carbonate rock areas to survive [1]. Thus, this method should be promoted globally.

### 4.4. Analysis of Limitation, Deficiency, and Uncertainty

Considerable basic data, including RS and measured data, are used in this study to calculate SFR. RS and reanalysis data with different spatial resolutions are also utilized. The data obtained through the method of adjacent data resampling are 8 km × 8 km (0.083°). The nearest neighbor method is simple and has small computational complexity but has poor visual effect. After sampling, discontinuing the image is outstanding, and the gray formation of the original image has high precision. We refer to the 1:400,000 soluble rock-type map and combine it with the 1:500,000 geological map when drawing the lithological boundary of China. The combination of strata with different scales may lead to errors.

In deeply negative assumptions, runoff in the study area does not present karstification. However, in pixels, the actual runoff depth of areas, such as Yuan, is difficult to determine through computations. Therefore, the method in [42,43] is adopted. The difference between precipitation and evaporation to replace the actual runoff depth may have a certain deviation between the two processes. Therefore, this method has certain limitations.

In the current study, the MPD method is used to calculate the dissolution rate. karst action in an open karst system may not reach the solution equilibrium. Therefore, the calculation result will deviate from the actual dissolution rate. In analyzing the correlation between *SFR* of different lithology and ecohydrological factors, the characteristics of seasonal changes may be ignored. Therefore, this work must be performed in a future study.

## 5. Conclusions

In this study, based on multi-source data and geospatial technology, we have used the MPD method and *SFR* calculation model to estimate the dissolution rate and *SFR* in karst areas of China during the period of 1983–2015, analyzed their spatial diversity and temporal variation, and reevaluated the risk of soil erosion. The following main conclusions were drawn: (1) Geospatial technology, combined with chemical thermodynamics theory, created the key link of geographical science and experimental chemistry, which can play an important role in the spatial calculation of rock weathering and soil formation rate. The calculated results were basically consistent with the actual monitoring of long-term positioning, which proved the practicability and extensibility of the method. (2) The dissolution rate of carbonate rock ranges between 0 and 106 mm/ka and has an average of 22.51 mm/ka. The *SFR* ranges between 10 and 134.93 t km$^{-2}$ yr$^{-1}$ and has an average of 18.59 t km$^{-2}$ yr$^{-1}$. A total of 86.44% of the study area has an *SFR* below 15 t km$^{-2}$ yr$^{-1}$, and most of the areas were distributed at HC. The risk of soil erosion in karst areas of China was reassessed, and the high erosion risk areas and ecological safety areas were corrected; we found that the former was nearly three times higher than the latter.

In conclusion, the contribution of this study is to solve the problem that traditional positioning monitoring is difficult to realize *SFR* spatialization, to promote the application of geospatial data and technology in new fields such as rock weathering and soil formation, and to provide an example and a new perspective for international counterparts to carry out relevant research.

**Author Contributions:** Conceptualization, S.W. and X.B.; methodology, G.L. and Q.L.; validation, Y.Y., and Q.L.; formal analysis, Y.T.; data curation, Z.H.; writing—original draft preparation, Q.L.; writing—review and editing, Q.L., Y.T. and X.S.; visualization, S.T. and Q.L. All authors have read and agreed to the published version of the manuscript.

**Funding:** The research was funded by United Fund of karst Science Research Center [grant numbers U1612441], National Key Research & Development Program of China [grant numbers 2016YFC0502102 & 2016YFC0502300], "Western light" Talent Training Plan of Chinese Academy of Sciences [grant number Class A 2018], Science and Technology Services Network Initiative Plan [grant number KFJ-STS-ZDTP-036], International Cooperation Agency International Partnership Program [grant numbers 132852KYSB20170029, 2014-3], Guizhou High-level Innovative Talent Training Program "Ten" Level Talents Program [grant number 2016–5648], National Natural Science Foundation of China [grant numbers 41571130074 & 41571130042], Science and Technology Plan of Guizhou Province of China [grant number 2017–2966].

**Acknowledgments:** We thank the Global Land Data Assimilation System (GLDAS), European Centre for Medium-Range Weather Forecasts, Global Inventory Modeling and Mapping Studies, GEMS-GLORI world river discharge database, China Meteorological information center and geo-engineering investigation institute of Guizhou have provided us for the study's data.

**Conflicts of Interest:** The author(s) declared no potential conflicts of interest with respect to the research, authorship, and/or publication of this article.

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
