# Peer review of "Change Detection of Soil Formation Rate in Space and Time Based on Multi Source Data and Geospatial Analysis Techniques"

_remotesensing, doi:10.3390/rs12010121_

Round 1

Reviewer 1 Report

Thank you for the thorough consideration of my comments, and the excellent additions to the manuscript. I have suggested some additional modifications that can result in substantial improvement in your article.
Please find my comments in the attached pdf.

Reviewer 2 Report

Dear Authors,

I have read the paper "Change detection of soil formation rate in space and time based on remote sensing applications" written by Li et collab., paper which was resubmitted for publishing in Remote Sensing Journal. The results of this new re-calculation/re-assessment of SFR at a whole scale of China will be appreciate by the scientific community. Comparing this new manuscript with the original one, I found good steps regarding RS databases and sources used. I suggest to the authors to rewrite the chapter 2.2., according to its title ... "and pretreatment", in order to make the method available and reproducible by the future readers.

Best regards

Reviewer 3 Report

The resubmitted manuscript is clearer and easier to understand. I believe that the applied methodology is sound and interesting and the obtained results are interesting as well.

However, a significant issue is that the title, the abstract and the conclusions are somehow misleading as RS techniques have not a central role in this study. The only relationship with RS is the use of spatial data from global databases that are based on the analysis of RS data. This fact does not reduce the merit of this study; however, the title the abstract, the conclusions etc. should be revised in order to represent the actual contents of this study. I provided some suggestions in my detailed comments. E.g. the title can be changed to: “Change detection of soil formation rate in space and time based on multi source data and geospatial analysis techniques.”

The language of the manuscript isn’t too bad, but some corrections are needed.

Please also see my detailed comments included in the commented pdf manuscript file.

Round 2

Reviewer 1 Report

Thank you for the thorough consideration of my comments, and the excellent additions to the manuscript.

This manuscript is a resubmission of an earlier submission. The following is a list of the peer review reports and author responses from that submission.

Round 1

Reviewer 1 Report

General Comments:

The title of the manuscript (MS) deals with Change detection of soil formation rate in space and time based on remote sensing applications. Generally, I found this to be an interesting and generally well-written manuscript. I have a number of relatively easily addressed comments detailed below.

Specific Comments:

Line 29: Delete "And". Line 75: you mentioned that a long-term high-resolution data of ecometeorological RS and monitoring were used, the resolution of the data should be mentioned here and in the section of “2. Data source and pretreatment Line 126: The "Methods" section, this section is one of the most important parts of a scientific manuscript and its aim is to give the reader all the necessary details to replicate the study. I strongly recommend to the authors devise a flowchart that depicts the steps that they have processed in this study. Lines 136, 137, 145, 146, 147, 148, 152, 153, 156, 161, 176, 189, and 204: All the equations from 1-13 here were added as images, and that made them unreadable/invisible (low quality especially when printing out the article). Therefore, all these equations must be re-rewritten using Insert/Equation in Microsoft office. Line 210: In the Results, section 3.1. "Diversity of dissolution rate and its evolutionary rule". I suggest that adding a few sentences to describe what this section is about. Line 233: “Figure 2. (b) of the average annual dissolution rate of 235 Karst in China during 1983~2010”. I suggest moving the legend outside of the figure and increase the font size, with this current size, I can hardly read it. Line 271: "Figure 4. (b) in 273 the Karst area of China during 1983~2010". I suggest moving the legend outside of the figure and increase the font size, with this current size, I can hardly read it. Line 326: "Figure 7. (b) SFR in China and ecohydrological factors, (c) SFR of CA 330 and ecohydrological factors, (d) SFR of CI and ecohydrological factors, (e) SFR of HD and ecohydrological 331 factors, (f) SFR of HDL and ecohydrological factors, (g) SFR of HL and ecohydrological factors." In all these statistics figures, the font size must be increased, with this current size, I could not see most of the numbers. Or they can be added as an appendix to the article. Line 335: In the Discussion, section 3.1. "4.1. Results verification and comparison with similar studies". I suggest that adding a few sentences to describe what this section is about.

Finally, and most importantly in the "Discussion" section, the authors should talk about the applicability of this method to another case study abroad. What are the mandatory prerequisites? Is it applicable to worldwide?

Reviewer 2 Report

Dear Authors,

I have read the paper "Change detection of soil formation rate in space and time based on remote sensing applications" submitted for publishing in Remote Sensing Journal. I am sure that the results of this new re-calculation/re-assessment of SFR at a whole scale of China will be appreciate by the scientific community. But, here appear some major issues: (i) can the title of the paper be supported by the content of the paper? (ii) remote sensing applications have been totally substituted by the paragraph 2.2. Data source and pretreatment (I really cannot find it) - rows 107 to 125; the risk reassessment of soil erosion is a much more complex problem than the authors addresses in the paragraph 4.2. (rows 363 to 387). I suggest to the authors to change the title of the paper and (consulting the editors) to find a most suitable journal to publish these results. In my opinion, not RS techniques and applications is the key of this paper (just a spatial database for the whole China). Otherwise, this can be seen in the Abstract and Introduction parts: here the authors describe what they really have carried out -  RS approaches are almost completely missing. Other specific suggestions/corrections can be find in the *.pdf file attached.

Reviewer 3 Report

The results of the study are interesting. However, the language is very confusing and the methodology isn’t presented adequately (actually only some fragments of what was done are presented). Accordingly, it is not possible to make a sufficient (in more depth) evaluation of the study. Furthermore, in the manuscript, remote sensing is only mentioned in the title, in the concluding phrase of the abstract, and the conclusions.

Specific Comments:

Title: There is no information about remote sensing in the manuscript. Lines 37-39: There isn't any hind about using remote sensing in the methodology and in the results of this manuscript. RS is only mentioned in the title the concluding phrase of the abstract and the conclusions. The language is very confusing in the entire manuscript. Please also check the style of the citations. Lines 77-82: The objectives are very confusing. The equations are very difficult to read. The methodology section is incomplete. The various equations and individual methods are described but there is no information about how all these are linked together, what is calculated with what data, what are the various steps of the calculation procedure and which is the ultimate goal. Furthermore, there is no information about how the various parameters were estimated. There is also no information about the geospatial analysis methods used as well as the corresponding tools and the spatial data characteristics. Figure 7. The figures 7b to g are very small and difficult to read. Lines 336 -362. Is the comparison with other studies a proper verification? Lines 352-353: Which scholars? Are there any studies to cite? Lines 376-378: Please justify this. How the erosion rate is a justification for SFR? Lines 391-396 (limitations): All this information is mentioned here for the first time. Lines 399-403 (limitations): All this information is also mentioned here for the first time.

Based on the above I have to suggest a major revision. However, the revisions needed are really major and many parts should be rewritten (especially the methodology section). A more in-depth evaluation of the revised manuscript is required, given that the revised manuscript will be clearer and more comprehensive.